# Analysis on personnel costs and working time for implementing a more person-centred care approach: a case study with embedded units in a Swedish region

Hanna Gyllensten [1,2] Malin Tistad,[3,4] Helena Fridberg,[3] Lars Wallin[1,3]

¹Institute of Health and Care Sciences, University of Gothenburg, Gothenburg, Sweden
²University of Gothenburg Centre for Person-Centred Care, Gothenburg, Sweden
³School of Health and Welfare, Dalarna University, Falun, Sweden
⁴Department of Neurobiology, Care Sciences and Society, Karolinska Institute, Stockholm, Sweden

**Correspondence to**
Hanna Gyllensten;
hanna.gyllensten@gu.se

## ABSTRACT

**Objectives** Our aim was to describe the time and costs used during the implementation of a more person-centred care (PCC) approach as part of ordinary practice.

**Design** A case study with embedded units.

**Setting** Region Dalarna, Sweden.

**Participants** The Department for Development (DD) staff who provided a central support function in the implementation and six healthcare units: nephrology, two geriatric care and rehabilitation units, two psychiatry units and primary care.

**Interventions** More PCC.

**Primary and secondary outcome measures** Working days and related salary costs reported by categories indicating costs for implementation strategies, service delivery, and research/development costs.

**Results** The healthcare units logged on average 5.5 working days per staff member. In the healthcare units, 6%–57% of the time reported was used for implementation strategies, 40%–90% for service delivery and 2%–12% for research/development. Of the time reported by the DD, 88% was assigned to implementation strategies. Costs associated with reported time indicated 23% of costs for this implementation occurred in the DD. Using the budgeted cost, this proportion increased to 48%. The budget for the DD corresponded to SEK 2.30 per citizen per year and 0.009% of the total healthcare budget of the region.

**Conclusions** The study found that a large part of resources used for this implementation of more PCC occurred in the DD, although at least half of the costs occurred in the healthcare units. Moreover, the cost of providing a central support function corresponds to a tiny proportion of the total health budget.

## INTRODUCTION

Healthcare systems in many countries experience increasing economic demands, both through the development of new technologies and treatments and through a changing age distribution in the population resulting in more people with multiple chronic conditions. As in many countries,[1 2] legislators and

### STRENGTHS AND LIMITATIONS OF THIS STUDY

⇒ The study was conducted by an independent research body that included researchers from different disciplines.
⇒ The included healthcare units were given multiple opportunities to clarify and correct their records.
⇒ The staff had no time set off for the logbooks and thus often filled in the logbooks retroactively, potentially resulting in recall bias.
⇒ Most of the observed changes were due to other factors than the implementation of a more person-centred care, due to parallel changes in these units.

healthcare organisations in Sweden have put person-centred care (PCC) high on the development agenda,[3] and management-control efforts are increasingly aimed at implementing PCC.

PCC acknowledges and endorses every person's resources, interests and needs, comprising shared responsibility and power, as well as coordinated care and treatment.[4–6] PCC strives towards a meaningful life, which differs from similar concepts (eg, PCC) that focus on functional life.[7] It is related to the integrated people-centred health services promoted by the WHO,[8] although without the community perspective embedded in the framework. PCC has been promoted to address patient dissatisfaction with healthcare access and delivery[9] and as a potentially cost-saving or cost containing measure through more effective use of resources.[10] Several studies suggest cost-saving as one argument for PCC.[10 11] Thus, PCC is expected to both improve care quality and contain costs.[12]

However, the knowledge about costs associated with introducing a more PCC is limited and scattered.[13] Some studies have included training costs in intervention costs[14 15] or indicated costs of transferring staff between

organisational units,[16] but to the best of our knowledge, its implementation costs have not been reported. Regardless of the little knowledge of costs associated with its implementation, PCC has a growing impact on the healthcare industry in many countries.[1 2] Overlooking such costs, however, implies that the time added for this implementation is minimal and can be ignored,[17] an assumption that to some extent contradicts previous findings that implementation of PCC was associated with increased job strain.[18]

Thus, this study aimed to describe the time and costs used during the implementation of a more PCC approach as part of ordinary practice. As far as we know, the current study is one of a few reporting health economic aspects of a more PCC approach in ordinary practice, that is, not as part of an intervention study.

## MATERIALS AND METHODS

Region Dalarna decided in 2015 to promote a more PCC approach throughout the health system. The implementation process was initiated and managed by the healthcare organisation Region Dalarna as an essential part of continuous quality improvement. The *Implementing PCC: Process evaluation of strategies, leadership and health economy* (IMPROVE) project was conducted in parallel by university researchers. The process evaluation focused on the implementation process rather than PCC (the innovation being implemented). This study thus reports on the resources used for implementing a more PCC approach in one Swedish region, using data from a case study with seven embedded units (*study protocol available as Supporting information*). Previous publications from the IMPROVE project have investigated how patients' perception of PCC can be measured,[19] how the concept of PCC was perceived by healthcare staff,[20] how the concept of PCC was operationalised by the units,[21] the used implementation strategies[22] and congruence of managers' perceptions and understanding of PCC across organisational levels.[23]

## SETTING

Region Dalarna is situated in the middle of Sweden. The region covers 6% of the Swedish land area and contains 3% of its population. Four hospitals and 25 healthcare clinics provides public healthcare for the region's population.[24] The Swedish health system provides universal coverage for all legal residents, and on some premises, also to visitors from other parts of the European Union, asylum seekers and undocumented immigrants. The health system is divided based on geographic area into 21 regions providing healthcare services and 290 municipalities providing care for the elderly and disabled. Approximately 84% of all health expenditures were tax-based in 2016, with care prioritised based on an ethical platform including the principles of human dignity, needs and solidarity and cost-effectiveness, in that order.[25]

## The innovation

The implementation of more PCC was conducted as part of the region's work towards efficient healthcare practices, in parallel with their 'structure and change work' that included projects related to priority setting and resource allocation in the regional healthcare system. The vision was to put equal emphasis on the patient and professional perspective throughout the care process. The approach chosen was based on the Gothenburg model of PCC,[5 26] which has been shown in clinical trials to be cost-effective for several care settings and patient groups.[10 27–29] The main feature of the model is its focus on the partnership between the patient and the healthcare provider built during the cocreation of a written health plan.[5] It has previously been reported that, although some ambiguity remained in their description, core practices related to all three cornerstones of PCC—creating, safeguarding and documenting the partnership—were identified in all the embedded units.[20] Important practices introduced in the units were routines to elucidate the patient narrative during admission or throughout the care pathway, and use of communication techniques such as motivational interviewing.

## The implementation process

As part of the implementation process, changes were made to the region's micro, meso and macro levels. These changes included the commission of staff in the regions Department for Development (hereafter called the DD; time committed to the project corresponding to 80% of a full-time employment), as well as a budget for expenses associated with the implementation. The DD staff assigned to lead this process were engaged as operational support provided centrally from the region to the included healthcare units. Among the DD tasks were to organise learning seminars and support the staff at the healthcare units during the implementation process. The participating healthcare units chose their representatives to participate in the learning seminars. Thereafter, each healthcare unit was expected to manage its implementation process. There was no joint implementation support programme other than the learning seminars, but the DD could provide support on-demand. The implementation strategies used have previously been described to mainly fall in two clusters, that is, train and educate stakeholder, as well as Develop stakeholder interrelationships.[22]

## The process evaluation

Interventions or programmes directed to changing healthcare provision, such as the implementation of PCC, are often described as complex interventions.[30] Complex interventions are interventions that contain several interacting components. This interaction can include elements tailored to each participant (eg, patients or healthcare staff), sometimes with varying outcomes and goals. In this study, complexity included differences in the training of participating staff, differences in organisational structures among units and each unit defining on their own how an

increase in PCC would be interpreted and implemented. It has been recommended that research into the use and effects of complex interventions address the complexity involved.[31] This can be done through process evaluations that provide an understanding of how the innovation (in this case PCC) is implemented.[32] In a conceptual model for implementation research, Proctor and colleagues elaborated on a number of implementation outcomes relevant for such an evaluation, including economic aspects of implementation.[33]

The process evaluation was conducted six healthcare units that participated in the first round of learning seminars and the DD. All units consented to participate in the evaluation. The healthcare units included were specialised in nephrology, geriatric care and rehabilitation, psychiatry, and primary care.

Costs related to the implementation included staff and the DD's costs, the training of staff and any support provided locally in the healthcare units. The study examines resources spent on making this change in healthcare but does not include any control units. Therefore, the study approach is in line with an observational study of a natural experiment in a real-world setting; that is, the research team has no control over the intervention, there is no control condition and the knowledge and availability of data for evaluation are partial.[34]

### Data collection

Data for this study were collected through logbooks, including date and type of activity, how much time each activity took (recorded as either minutes, hours or workdays) and for how many people, as well as information about who was involved. Each unit selected one person in a leading position in the implementation to complete the logbooks, either as each activity was conducted or retrospectively. These persons received the same instruction in how to use the logbooks. However, in order not to influence the choice of strategies, we did not give any guidance regarding taxonomies that could have been used to choose or describe the activities carried out to support the implementation. Reporters were encouraged to report short and often reoccurring activities (eg, discussions about implementation between colleagues

during the workday) as weekly estimates. The persons responsible for the logbooks were encouraged to report on a weekly to monthly basis but some had difficulties adhering to this recommendation due to a high workload and were instead encouraged to use their calendars on a half year basis to track their activities. In some instances, representatives from the research group met with the person responsible for the logbook and assisted to fill out the logs.

Information from the logbooks was used to identify which activities were perceived by personnel units to be related to implementing a more PCC approach and estimations of the time used for this implementation. Only time was reported in the logbooks, no equipment or other expenses were tracked.

In addition, units were asked to provide suggestions (hereafter called unit-specific measures) to evaluate the economic impact of implementing a more PCC approach (table 1), that leaders in these units viewed as important for understanding the changes in practice induced by PCC. Relevant data on these proposed outcomes were collected from each unit retrospectively, and the units were asked to comment on any potential time trends in the data.

Data collection was planned to cover a 3 year study period (June 2016–May 2019), but logbook data continued into autumn of 2019 for the psychiatric units and the DD due to delays in previously planned activities. Activities conducted before the ethics approval (in 2017) were filled in retrospectively.

### Resource use and cost estimation

In analysing implementation programmes, four categories of costs have previously been suggested in the literature:[35] (1) costs for executing implementation strategies, (2) excess costs for service delivery as it changes, (3) opportunity costs to providers and patients and (4) research/development costs. These categories were used retrospectively by the research group to categorise activities reported in the logbooks. In this study, costs for executing implementation strategies are mainly directed towards centrally organised processes (eg, seminars). In contrast, the costs for service delivery included activities

**Table 1** Preplanned unit-specific measures of resource use (for study protocol, see online supplemental file 1)

| Unit | | Planned reporting |
|---|---|---|
| Nephrology/dialysis unit* | | Distribution between haemodialysis, assisted peritoneal dialysis and unassisted peritoneal dialysis; work hours (adjusted for the number of patients) |
| Primary care* | | Number of patients listed; number of visits per patient |
| Geriatric care and rehabilitation | Unit 1: | Length of stay |
| | Unit 2: | Work hours*; number of patients; work environment follow-up* |
| Psychiatrics† | Unit 1: | Length of stay (considering overcrowding) |
| | Unit 2: | Length of stay and readmissions (considering overcrowding) |

*Not reported (deviation from planned analyses).
†These two units were merged during a large part of the study, and unit-specific measures were thus reported as a combined unit.

**Table 2** Resource use reported in logbooks by categories of costs and organisational units

| Organisational unit (approx. employees) | Total | Cost categories* | | |
| | Working days† (per employee) | Implementation strategies | Service delivery | Research/development |
| | | Working days† (% of total) | Working days† (% of total) | Working days† (% of total) |
| --- | --- | --- | --- | --- |
| Geriatric unit 1 (50) | 185 (4) | 19 (10) | 161 (87) | 5 (3) |
| Geriatric unit 2 (50) | 167 (3) | 95 (57) | 67 (40) | 5 (3) |
| Nephrology unit (40) | 162 (4) | 68 (42) | 74 (46) | 20 (12) |
| Primary care unit (70) | 275 (4) | 44 (16) | 212 (77) | 19 (7) |
| Psychiatric unit 1 (20) | 263 (13) | 15 (6) | 236 (90) | 12 (5) |
| Psychiatric unit 2 (20) | 95 (5) | 19 (20) | 74 (78) | 2 (2) |
| DD (18) | 267 (–)‡ | 234 (88) | 5 (2) | 28 (10) |

Numbers are rounded.
*Developed from previously reported cost categories in implementation programmes: (1) costs for executing implementation strategies, (2) excess costs for service delivery as implementation/service changes, (3) opportunity costs to providers and patients and (4) research/development costs.[35] No account was taken for changes in costs for direct patient care (category 3).
†Assuming 8 hour working days.
‡Time per employee was not calculated for the Department of Development as the time used was mainly spent by the two persons engaged in providing operational support.
DD, Department for Development.

the units used in operationalising PCC in ordinary practice. This study did not reflect foregone opportunity costs in the care of patients because data collection was not designed for patient-level follow-up. However, it needs to be acknowledged that all resource use in the health system can potentially have an opportunity cost related to an alternative use of existing resources. Activities reported in logbooks were first categorised inductively to identify less aggregated types of resource use and, after that, deductively according to the above categories. Conversions were made assuming 8 hour workdays and 46 full weeks of work each year.

Costs for used resources were calculated based on the time used for each activity reported in the logbooks. The corresponding costs to the health system in 2019 values were calculated by multiplying the time spent by wage and the related mandatory and negotiated social insurance contribution (37.14%[36]). The mean wage for a nurse employed by a Swedish Region was SEK 34 100,[37] which was used as an approximation for the wage of staff in the healthcare units as we seldom knew the distribution of staff categories in logbook recordings (ie, SEK 2440 per day working, after adjusting for holidays and social insurance contributions, calculated as SEK 34 100×1.3714×12 months divided by 46 weeks working and 5 days per week). The approximate wage for staff employed in the DD was SEK 40–45 000 reported by the development leaders, which in the analysis was approximated to SEK 43 000 (resulting in SEK 3077 per day working). Costs for the DD were also calculated using information from the regional budget documentation for this function, including a set budget for each year during the study. The set budget, for instance, was to cover 80% of one full-time employee. The budget was further contextualised using the total population of Region Dalarna (approximately 281 000 inhabitants) and the total healthcare spending during the study period. Year-end reports from the region reported costs of SEK 8985–11 106 million[38] during the years 2016–2018. Costs were analysed from the health systems perspective, including all costs to the care organisation, and expressed in 2019 value. No discounting was deemed necessary because costs were only reported descriptively.

### Statistical analyses
Time reported in the logbooks was reported descriptively by types of activity associated with implementing a more PCC approach (online supplemental file 2). The unit-specific outcomes were reported separately for each unit, using graphs to illustrate trends over time. The linear equation was used to indicate trends in the length of stay while survival analysis was used to examine time to readmission. Where applicable, analyses were adjusted for overcrowding during the initial hospitalisation. All analyses were conducted using Stata Statistical Software: Release 17.0. College Station, TX: StataCorp LLC.

### Patient and public involvement
This project did not include patient or public involvement in developing the research questions, design, conduct, choice of outcome measures, or recruitment.

### RESULTS
The time reported in logbooks was between 3 and 13 working days per staff member in the participating units (table 2), although the time spent was not equally

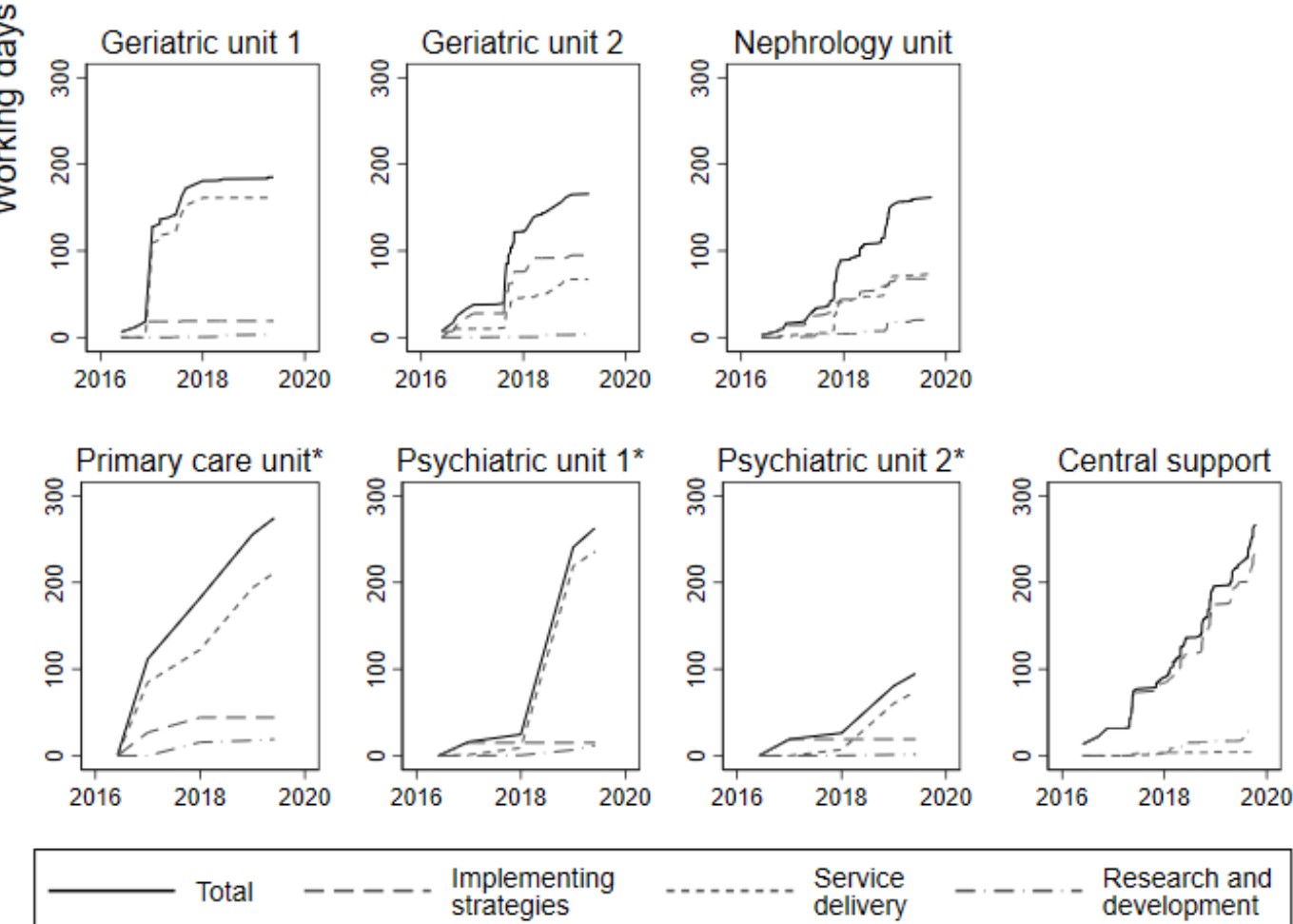

**Figure 1** Accumulation of working days for the Department for Development and the care units by categories of costs in implementation programmes. *Resource use analysed by year (other graphs are based on actual dates).

distributed among the teams in each unit (mean 5.5 working days; median 4 working days). In total, time reported in logbooks from the DD corresponded to 267 full days of work (account for work done by the development unit overall, including the time used by two assigned development leaders) and 95–275 full days of work per unit for the healthcare units, over the 3 year study period. The cumulative distribution of time reported in logbooks indicate that some units reported an equal workload associated with the implementation between years, while others reported a more varied pattern of workload (figure 1).

Activities categorised as being related to the implementation strategies (88% of time reported by the DD, 6%–57% for the other units) included planning and preparatory work for the learning seminars, as well as conducting/participating in those seminars. For the DD, service delivery (2% of their total reported time) was interpreted to include participation in regional decision making, administrative work, reporting to the region and collaboration with other organisations (eg, unions). For the units, it included IT solutions, reporting, educational activities, development of teams and care development (corresponding to 40%–90% of their reported time).

Research and development costs comprised interactions with the research team and external collaboration with other regions (10% of the time reported by the DD, 2%–12% for the other units). A more detailed list of the activities reported by the units and the distribution of reported time between activities are reported in eTable 1 (online supplemental table 1). Training of staff was a major part of this implementation programme. For some employees, the training included up to three learning seminars (7–8 hours/seminar, difference based on transportation), that is, implementation strategies. In most units, additional training sessions were added (for a selected group or all employees) to facilitate the implementation of a more PCC approach, such as training in communication techniques, that is, service delivery categorised under either care development or team development depending on the type of training chosen (online supplemental table 1).

In table 3, only salary costs for the healthcare units are listed, estimated based on an approximate wage per day of a nurse or DD staff, respectively. For the healthcare units, salary costs ranged from SEK 231 582 to SEK 669 922. For DD, the salary costs for activities reported in the logbooks were calculated to SEK 822 633 However,

**Table 3** Costs for time reported in logbooks by categories of costs and organisational units

| Organisational unit (approx. employees) | Total SEK (% of total cost for all units combined) | Cost categories* Implementation strategies SEK | Service delivery SEK | Research/development SEK |
|---|---|---|---|---|
| Geriatric unit 1 (50) | 451 673 (12) | 46 358 | 393 535 | 11 779 |
| Geriatric unit 2 (50) | 406 930 (11) | 231 638 | 163 778 | 11 514 |
| Nephrology unit (40) | 394 811 (11) | 165 761 | 179 638 | 49 413 |
| Primary care unit (70) | 669 922 (19) | 107 661 | 516 039 | 46 223 |
| Psychiatric unit 1 (20) | 642 001 (18) | 36 598 | 575 740 | 29 662 |
| Psychiatric unit 2 (20) | 231 582 (6) | 46 358 | 181 493 | 3731 |
| DD (18) | 822 633 (23) | 720 334 | 15 384 | 86 916 |
| Total cost | 3 619 552 (100) | 1 354 708 | 2 025 607 | 239 238 |

Numbers are rounded.
Costs include wages and social insurance contributions but excludes overhead costs (eg, facilities).
*Developed from previously reported cost categories in implementation programmes: (1) costs for executing implementation strategies, (2) excess costs for service delivery as implementation/service changes, (3) opportunity costs to providers and patients and (4) research/development costs.[35] No account was taken for changes in costs for direct patient care (category 3).
DD, Department for Development; SEK, Swedish krona.

the DD included both personnel (budgeted as 80% of one full-time employment over the whole period, corresponding to 552 working days (calculated as 80% of full-time over 3 years (with full time being 46 weeks per year))) and resources for training (seminars and workshops), development work and IT resources, as well as internal and external communication. For 2016, 2017, 2018 and 2019, the annual budget was SEK 500 000, SEK 600 000, SEK 600 000 and SEK 880 000, respectively. Thus, for the work organised by the DD during the period 2016–2019, the total budget was SEK 2 580 000, of which approximately SEK 1 698 342 were salary costs for the 80% employment to support the units, making it clear that the data collection through logbooks, SEK 822 633, did not capture all activities conducted by the DD (ie, the 80% of a full-time employment over the whole study period; SEK 1 698 342).

Based on the target population of Region Dalarna, the total cost of the DD (salaries and other expenses) corresponds to SEK 2.30 per citizen per year. Thus, the DD budget corresponded to 0.009% of the total healthcare budget over the studied period (June 2016–May 2019). The approximate exchange rate is SEK 10 ≈ EUR 1 (exchange rate in 2019 was mean SEK 10.5892 (range SEK 10.1874–10.9056) per EUR 1[39]), which means the total budget for the DD was approximately EUR 258 000 during 2016–2019.

Salary costs associated with time reported in logbooks indicated 23% of costs (SEK 822 633 of a total SEK 3 619 552, table 3) for this implementation occurred in the DD. Including the total budgeted funding for the DD (SEK 2 580 000), their proportion increased to 48% of the total cost.

### Unit-specific measures
The first geriatric unit online supplemental figure 1 and the psychiatric units (combined in online supplemental figure 2) reported a decrease in the average length of stay among their patient populations during the study period. Further examination of data from the psychiatric units showed that this trend was not explained by overcrowding (results not shown). Time to readmission within the first 10 months after discharge was similar between years in psychiatric units (combined in online supplemental figure 3). The apparent increase in length of stay in the psychiatric units (online supplemental figures 2,3) is affected by data availability and how the graph was created, with longer hospitalisations started before the end date of the data collection continuing into the second half of 2019 while no shorter stays are added.

The second geriatric unit reported a similar number of discharges across years (range 675–710), but refrained from providing further unit-specific data due to significant changes in the organisation (box 1). The nephrology and primary care units also refrained from delivering the planned outcome data due to other major changes in their work processes conducted during the same period as the implementation of a more PCC approach. Consequently, these data are only slightly related to the change under study. The primary care unit had approximately 14 000 listed patients throughout the study period.

### DISCUSSION
The healthcare units logged on average 5.5 working days per staff member for implementing a more PCC approach, but the number of days varied largely between units (range 3–13 working days). In the healthcare units,

## Box 1 Staff turnover and changing methods for prioritising patients: an example from geriatric care and rehabilitation (geriatric unit 2)

Initial contacts to discuss the evaluation of resource use were able to identify several aspects that could be relevant to follow during the development towards a more PCC approach, including work hours, number of patients, work environment follow-up and information that should be accessible through the administrative registers. These aspects were considered especially important due to the shortage of registered nurses.

When the project was nearing its end, new contacts were made. None (or very few) of the people involved in the project's launch remained in the organisation in 2019 due to changing roles or retirement. Concerns were expressed that they had not had time to actively work on the person-centeredness due to staff shortages and related downsizing of patient beds. The reduction in patient beds was described as having a budget for 18 patients but only staff enough to admit 10. They had handled the lack of nurses by changing from a registered nurse and an assistant nurse working in pairs to each nurse working with two assistants and transferring tasks to the physicians. Daily discussions were held to ensure that those in most need of the services provided by the geriatrics unit were cared for at the unit, and not moved to other sections of the hospital due to overcrowding.

Thus, it was concluded that it would not be relevant to evaluate any of the initial planned unit-specific outcomes given that the implementation process of a more PCC approach had been given a secondary role compared to other changes in the unit. In table 2, it can be seen that this unit used among the lowest number of working days per person of all included units and had the highest percentage (57%) of that work distributed to the initial planning and participation in learning seminars (cost category implementation strategies). Combined, the perceived secondary role of the implementation process and the discrepant time distribution compared with other included units can indicate the implementation process was not complete.

PCC, person-centred care.

6%–57% of the time reported in logbooks was assessed as being used for implementation strategies, 40%–90% for service delivery and 2%–12% for research/development. As expected, the distribution of time used by the DD staff differed considerably from that of the other participating units, with most of the logged time (88%) assigned to implementation strategies. While the time spent and salary costs associated with the implementation process were considerable, usually corresponding to at least 0.5–1 year of full-time employment per unit, the total cost was small compared with the entire healthcare budget. Although budgeting for this implementation was only available for the DD, at least half of the costs occurred in the other healthcare units. Unit-specific outcomes from three of the units showed no clear effect of the implementation, and in general, the healthcare units reported that other factors had affected their throughput more during this study period than the implementation of a more PCC.

To the best of our knowledge, this is the first study investigating the different components contributing to the time and costs spent on implementing a more PCC approach. Several studies have showcased how implementation costs should be measured,[35 40–42] but there is a shortage of studies measuring the costs from an implementation standpoint.[43 44] The strength of the study is that it was conducted by an independent research body that included researchers from different disciplines relevant to the interpretation of the results. Another strength was that the included units were given multiple opportunities to clarify and correct any oversights in their records (such as not writing the number of participants during a specific action) and potential misunderstandings regarding interpretation. Although the independence of the researchers was a strength, it also contributed to the main limitation of the study, the data collection process. The staff had no time set off for the logbooks and were not provided with extra time or staff to conduct the implementation process. Some reporters filled in large parts of the logbooks retroactively (not only 2016 data), potentially resulting in recall bias, and in some cases, one of our research group members (MT or HF) took part in this work to record previous activities. It should be noted that for one of the units with the highest working day estimate in relation to their staff, one of the researchers supported the staff member in filling out the logbooks which could indicate the staff member was reminded of more tasks having been conducted than would otherwise have been the case. However, the higher estimate could also be the result of this being one of two units that were merged during part of the study, thus making the division of time spent for each of these units difficult to distinguish.

While all unit-specific outcomes had been identified by each healthcare unit as important aspects to follow during this change process, most units chose in the end to not providing the data due to being more affected by other changes in the workplaces. For the units providing these data, there is still an assumption that most of the observed changes were due to other factors than the implementation of a more PCC. The reason for initiating the collection of unit-specific outcomes was to make the evaluation more relevant to the participating healthcare units and similar units elsewhere, and to ensure that the implementation of a more PCC approach was at least not associated with any large negative impact on patient throughput. However, due to other parallel changes in these units, it did not provide any conclusive results. The commitment shown by staff in the participating units is exemplified by them participating in the change process and providing materials and responses to questions during analysis, regardless of the lack of time provided for their participation. This commitment can also be discussed regarding the planned observational approach of the study. Here, it can be argued that by engaging in research and data collection, the staff involved in the implementation may have been affected (ie, through social desirability bias or being reminded of the implementation process by the researchers) and thus to a larger extent interacting with it. While initially intended, we did not have logbooks from leaders (chief executives) of this implementation

process at the main organisational level in the region, but that should have implied a small cost compared with that of changing care practice. It should also be acknowledged that the division of costs by categories is not self-evident. Costs for changes in service delivery could also to some extent be seen as costs for implementation strategies. For example, the initial learning seminars were assumed to be part of the implementation strategies. However, if the healthcare unit later decided that in their work to change practice, the staff needed special training (eg, in interprofessional rounds or motivational interviewing), we interpreted this as part of changing how service was delivered. An alternative interpretation would have been to see this training as part of an iterative process of implementation strategies conducted at different levels of the organisation, that is, the study distinguishes between centrally planned implementation strategies and strategies conducted by the healthcare delivery organisation.[45] When considering the limitations of the study, our estimated working days and costs should be interpreted with caution. These findings are only part of a picture that needs to be further developed in future studies and frameworks to assess the economic impact of implementation.

Using the cost components suggested by Wagner *et al*[40] as a basis, additional costs would include overhead costs (eg, facilities). Moreover, the inclusion of research and development costs should be broken down into what would be sunk costs (ie, one-time investments) versus costs for development and scaling up (such as communications within the region to support others) versus costs that are solely for research purposes (ie, the research interviews). However, it has been argued that the costs of research should be reported,[46] not to inform clinical practice but to assess the costs associated with evaluation. Here, we are only reporting on the costs for the region to participate in the research study. We hope our approach and findings might help others design similar studies to follow implementation processes more in depth.

Changes in healthcare can aim to either improve effectiveness, that is, save money while producing at least as much health as before, or increase care quality or a combination of the two.[47] Today, there is a growing body of evidence of improved patient outcomes and possibilities for cost savings by shifting to PCC from randomised controlled trials and quasi-experimental studies in Sweden and internationally.[10 28 29 48 49] In comparison, little is known about the 'hidden' costs of preparatory work, training and monitoring outcomes during implementation,[44] costs that are investigated in this study. In a recent systematic review of the literature, only six studies were identified that specified such costs. It has been argued that any implementation effort should be preceded by ex-ante modelling to compare the expected returns of implementing the intended change to the predicted costs of implementing the change.[35 43 50] However, this is not always the case, or the results are at least not available to the research community.

One crucial aspect of the studied change process was that it was made clear that healthcare units would not be provided with extra time or other resources for the implementation. This was reported by both the central regional organisation and the management in each healthcare unit, with staff shortage being the main reason. However, we fund that units implementing more PCC used a considerable amount of time for this implementation process. Thus, the time reported in logbooks could be interpreted to refer to the time that otherwise would have been used for other work in the healthcare units. Had the implementation of more PCC not been made, the reported time would still have been used in the units. A hypothetical comparator, an alternative intervention, could have used the same amount of time to implement some other change in practice, such as developing and implementing clinical guidelines or care paths (which thus would suggest the opportunity cost of this implementation process). Because several units had assigned quality developers or specialist nurse students with tasks associated with the implementation process, which equalled several working days spent per staff member, the used time likely offset other tasks otherwise conducted by the staff. In addition, staff in some settings expressed that it was time-consuming to provide PCC in the immediate time frame but it could potentially be beneficial later in the care process.[20] Together with recent reporting that increased person-centredness was associated with higher job strain,[18] it is likely that additional resources during the initial period had resulted in improved uptake.

It should also be noted that the possibility to influence resource use is probably influenced by the patient groups in each healthcare unit and to what extent work with these patients is already streamlined. Considering a patient group for which clinical guidelines determines the frequency of follow-up, there is less opportunity to change the number of visits and thus costs will be similar even if the care changes. If there is instead a patient group experiencing unmet needs and much acute unplanned care, it can be assumed that changing how patients experience their healthcare can change how many visits are needed.

While several of the healthcare units expressed that they had not completed the implementation within the study period, assessing its success need to be based on still ongoing studies of patients' experiences. However, the findings clearly demonstrate that there is a non-zero cost of implementing a more PCC approach; costs that should be acknowledged in future research and implementation processes. The study also points towards potential improvements in how to study implementation costs, through, for example, recurrent questionnaires instead of logbooks that are collected at the end of the study period. Furthermore, due to the reported high staff turnover, the costs for changes in service delivery may to some extent continue in training of new staff. Considering our findings in light of recent updates on the use of economic evaluation in implementation science to guide

decision-makers,[51 52] future studies should thus distinguish all costs associated with implementation science, including implementation costs, intervention costs and downstream costs[17] of a more PCC approach, as well as for other healthcare programmes.

## CONCLUSIONS

The study found that a large part of resources used for this implementation of more PCC occurred in the DD, although at least half of the costs occurred in the healthcare units. Our findings suggest that the main costs associated with implementing a more PCC approach in ordinary practice resulted from implementing strategies and service delivery. In contrast, research and development costs were small by comparison. Moreover, the cost of providing a central support function corresponds to a tiny proportion of the total health budget. While there are limitations in how the study was conducted, it clearly demonstrates a non-zero cost of implementing a more PCC approach, thus implicating that future research should capture costs. Not accounting for the added strain on healthcare units can result in delay or inability to implement the new care model.

**Acknowledgements**  We thank all contributing health care units. Thanks also to Dr Leslie Shaps for language editing.

**Contributors**  HG, MT, HF and LW contributed to the design of the work; HG, MT and HF contributed to the acquisition, analysis and interpretation of the data; HG drafted the manuscript while MT, HF and LW substantively contributed to the final manuscript; HG, MT, HF and LW all have approved the submitted version and agree both to be personally accountable for the author's contributions and to ensure that questions related to the accuracy or integrity of any part of the work, even ones in which the author was not personally involved, are appropriately investigated, resolved and the resolution documented in the literature. HG is the guarantor of the study.

**Funding**  The study was financed by the University of Gothenburg Centre for Person-Centred Care (GPCC) and Dalarna University. The funding body had no role in the design of the study, in collection, analysis and interpretation of data, or in writing the manuscript.

**Competing interests**  None declared.

**Patient and public involvement**  Patients and/or the public were not involved in the design, or conduct, or reporting or dissemination plans of this research.

**Patient consent for publication**  Not required.

**Ethics approval**  This study involves human participants and was approved by the regional ethics committee in Uppsala (approval number: 2017/195). Participants gave informed consent to participate in the study before taking part.

**Provenance and peer review**  Not commissioned; externally peer reviewed.

**Data availability statement**  All data relevant to the study are included in the article or uploaded as supplementary information. All relevant data are within the manuscript and its Supporting Information files (supplementary file S2).

**ORCID iD**
Hanna Gyllensten http://orcid.org/0000-0001-6890-5162

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
