## [Reviewer comments · BMJ Open]

ARTICLE DETAILS

TITLE (PROVISIONAL)	An analysis on personnel costs and working time for implementing a more person-centred care approach: A case study with embedded units in a Swedish region
AUTHORS	Gyllensten, Hanna; Tistad, Malin; Fridberg, Helena; Wallin, Lars

VERSION 1 – REVIEW

REVIEWER	Annabel Stierlin Ulm University, Institute of Epidemiology and Medical Biometry
REVIEW RETURNED	12-Jun-2023

GENERAL COMMENTS	Thank you for the opportunity to review this manuscript on costs associated with the implementation of a more person-centred care approach. The need for implementing person-centred care approaches in health systems is obvious and the objective to assess the associated costs for health systems is very important. Though, the validity of the data collected in this study is questionable. It might be a matter of reporting, but I got the impression that data quality is rather poor. It seems there were several factors impeding data collection, e. g. leading to retrospective data collection. In view of potential competing interests, lack of documentation time, and interindividual heterogeneity in documentation styles, an observational method by independent, trained raters would be recommended, instead of self-reported logbooks. For this reason, it seems very difficult to draw conclusions based on these results.
---

REVIEWER	Marina Di Giacomo Università degli Studi di Torino, Economics, Social Science, Mathematics and Statistics
REVIEW RETURNED	18-Jul-2023

GENERAL COMMENTS	The paper aims to estimate the costs of implementation of a PCC approach in six healthcare units in a Swedish region. The data come from logbooks filled in by the units' responsables. These logbooks cover May 2016-June 2019, i.e., six months before and 18 months after implementing the new approach. The paper is interesting, and it tackles an important issue. Here are some suggestions that may improve the readability of the work: - On page 7, rows 44 and subsequent: It would be helpful for more information about the DD: what is its role? Did it also exist before the PCC program? Is it a completely new or existing department where 80% of the staff is now devoted to the PCC program?- On page 9, lines 40-48: I think the timeframe should be more
---

	carefully explained in the data collection section. Carefully explain when the new PCC approach started and when data started to be collected. Also, it is not clear the statement 'Activities conducted before the study started (ethics approval in 2017) were filled in retrospectively'. When did the study start? Which is the exact period for the retrospective collection of data? - On page 10, line 37: the statement "four categories of costs have been suggested" needs some clarification. Who suggested these categories? Why? And related to this: what exactly are the instructions given to the unit's leaders in charge of the logbook completion about categorizing the different costs? - Linked to the last comment above: it is crucial to know whether or not each unit leader in charge of the log book received any particular guidelines for compiling the logbook and if these guidelines were identical across the different units. It is quite striking in Table 2 that the two geriatric units differ so much in their 'cost structure.' Also, the two psychiatric units are quite different in their cost structures. - On page 12, line 30: 'and 77-281 full days of work per unit for the health care units'. I cannot find these figures in Table 2. Probably, it should be 'and 95-275 full days of work per unit for the health care units'. Please, carefully check all the figures in the text and in the Tables. - On page 12, line 41. I do not completely agree with the statement, 'Moreover, activities interpreted as associated with the implementation strategies were in most units corresponding to the majority of time reported throughout the study period.' This is true for only one unit (geriatric unit 2) and the DD. In contrast, based on results from Table 2, I think activities within the 'service delivery category' take most of the time in most units. - On page 13, I would add some comments for the activities in the service delivery categories based on STable 1. - On page 14: the comments to Table 3 are not very clear and should probably be re-written. Here are my suggestions. First, comment upon the costs of each unit, reminding the reader that these costs are computed using the equivalence SEK 2440 per working day, where SEK 2440 corresponds to the daily wage of a nurse. Then you can comment upon the DD costs. The situation is more complicated here, and the authors should guide the reader through the discussion. In Table 3, only personnel costs are reported (and the equivalence is different from those of other units). Moreover, additional costs are available for the DD, etc... - In SFigure 2: what did it happen at the end of 2019? It seems there is a sharp increase in the length of stay. - The paper's title should emphasize that only some costs (personnel costs) or, better, the role of working time in a PCC approach implementation are considered. For example: 'An analysis on personnel costs and working time for implementing of a more person-centred care approach: the IMPROVE project,' or something along these lines.
--	--

VERSION 1 – AUTHOR RESPONSE

Reviewer 1, comments to the Author:

Thank you for the opportunity to review this manuscript on costs associated with the implementation of a more person-centred care approach. The need for implementing person-centred care approaches in health systems is obvious and the objective to assess the associated costs for health systems is very important. Though, the validity of the data collected in this study is questionable. It might be a matter of reporting, but I got the impression that data quality is rather poor. It seems there were several factors impeding data collection, e. g. leading to retrospective data collection. In view of potential competing interests, lack of documentation time, and interindividual heterogeneity in documentation styles, an observational method by independent, trained raters would be recommended, instead of self-reported logbooks. For this reason, it seems very difficult to draw conclusions based on these results.

Response: Thank you for your comments. We agree that there are some limitations to the methods used, which is brought up in the discussion.

Reviewer 2, comments to the Author:

The paper aims to estimate the costs of implementation of a PCC approach in six healthcare units in a Swedish region. The data come from logbooks filled in by the units' responsables. These logbooks cover May 2016-June 2019, i.e., six months before and 18 months after implementing the new approach.

The paper is interesting, and it tackles an important issue. Here are some suggestions that may improve the readability of the work:

Response: Thank you for your careful review of the manuscript.

- On page 7, rows 44 and subsequent: It would be helpful for more information about the DD: what is its role? Did it also exist before the PCC program? Is it a completely new or existing department where 80% of the staff is now devoted to the PCC program?

Response: We agree that this was not clearly described, we have now clarified in (currently) page 6 (heading The implementation process) that it was staff from the hospitals own department that were engaged, and that their total commitment to the project corresponded to 80% of a full-time employment, not 80% of all staff in that unit.

- On page 9, lines 40-48: I think the timeframe should be more carefully explained in the data collection section. Carefully explain when the new PCC approach started and when data started to be collected. Also, it is not clear the statement 'Activities conducted before the study started (ethics approval in 2017) were filled in retrospectively'. When did the study start? Which is the exact period for the retrospective collection of data?

Response: The period (studied period v data collection) has been clarified.

- On page 10, line 37: the statement "four categories of costs have been suggested" needs some clarification. Who suggested these categories? Why? And related to this: what exactly are the instructions given to the unit's leaders in charge of the logbook completion about categorizing the different costs?

Response: We moved the reference to make it clearer that these categories were from a specific publication (by Hoomans and Severens) and clarified that the categories were used by the research group to categorize reported activities, not by the staff in each unit.

- Linked to the last comment above: it is crucial to know whether or not each unit leader in charge of the log book received any particular guidelines for compiling the logbook and if these guidelines were identical across the different units. It is quite striking in Table 2 that the two geriatric units differ so much in their 'cost structure.' Also, the two psychiatric units are quite different in their cost structures.

Response: We have added information on page 7 on the instructions (which were the same for all units) that were provided for completing the logbooks.

With regards to the differences in cost structure between similar units, it should be noted that each unit were free to plan their implementation process and the activities associated with this process; therefore, it differed substantially between units. We believe that exactly this is among the more important results in this project, how large of a difference it is and how challenging it was to get an implementation process underway for units, while at the same time struggling with lack of and changing staff and leadership. This is very different from how resource use looks in many researcher-led intervention study, where the implementation cost sometimes even can be viewed as negligent (even if it's just as high as what we see here) since it will be very small compared to all the healthcare resource use among future patient groups. We are trying to describe that process, how the resources are used or at least is perceived to be used, rather than comparing it between alternative uses, since the challenge to include the new way of working in usual care in the now will remain even if its potentially offset by future savings.

- On page 12, line 30: 'and 77-281 full days of work per unit for the health care units'. I cannot find these figures in Table 2. Probably, it should be 'and 95-275 full days of work per unit for the health care units'. Please, carefully check all the figures in the text and in the Tables.

Response: You are absolutely correct; we have checked the numbers again throughout the manuscript. Thank you for pointing this out.

- On page 12, line 41. I do not completely agree with the statement, 'Moreover, activities interpreted as associated with the implementation strategies were in most units corresponding to the majority of time reported throughout the study period.' This is true for only one unit (geriatric unit 2) and the DD. In contrast, based on results from Table 2, I think activities within the 'service delivery category' take most of the time in most units.

Response: Agree, the statement has been removed.

- On page 13, I would add some comments for the activities in the service delivery categories based on STable 1.

Response: This has been elaborated.

- On page 14: the comments to Table 3 are not very clear and should probably be re-written. Here are my suggestions. First, comment upon the costs of each unit, reminding the reader that these costs are computed using the equivalence SEK 2440 per working day, where SEK 2440 corresponds to the daily wage of a nurse. Then you can comment upon the DD costs. The situation is more complicated here, and the authors should guide the reader through the discussion. In Table 3, only personnel costs are reported (and the equivalence is different from those of other units). Moreover, additional costs are available for the DD, etc...

Response: Thanks for pointing this out, the paragraph has been rewritten and divided into two for ease of reading.

- In SFigure 2: what did it happen at the end of 2019? It seems there is a sharp increase in the length of stay.

Response: This has to do with the registration of hospitalizations, we asked to get information about all hospitalizations starting before a certain date so the only ones that are seen during that period are those longer hospitalizations (while many of the shorter ones during those last months in the graph started after our data collection really ended). This has now been clarified better in the results section, heading Unit specific measures.

- The paper's title should emphasize that only some costs (personnel costs) or, better, the role of working time in a PCC approach implementation are considered. For example: 'An analysis on

personnel costs and working time for implementing of a more person-centred care approach: the IMPROVE project,' or something along these lines.

Response: Thank you for the suggestion, we have worked this into the adjusted title.